# Effect of Small-Sided Games with and without the Offside Rule on Young Soccer Players: Reliability of Physiological Demands

**DOI:** 10.3390/ijerph191710544

**Published:** 2022-08-24

**Authors:** Igor Junio Oliveira Custódio, Renan Dos Santos, Rafael de Oliveira Ildefonso, André Andrade, Rodrigo Diniz, Gustavo Peixoto, Sarah Bredt, Gibson Moreira Praça, Mauro Heleno Chagas

**Affiliations:** School of Physical Education, Physiotherapy and Occupational Therapy, Federal University of Minas Gerais, Av. Antônio Carlos, 6627, CEP 31270-901, Belo Horizonte 31270-901, Brazil

**Keywords:** small-sided games, task constraints, physiological demand, young soccer player, peak heart rate, offside rule, reliability

## Abstract

This study aimed to compare the physiological demand between three vs. three small-sided games (SSGs) with (3vs.3_WITH_) and without (3vs.3_WITHOUT_) the offside rule, as well as the within- and between-session reliability of this demand. Twenty-four U-17 soccer athletes performed various three vs. three (plus goalkeepers) SSGs with and without the offside rule. The data collection was performed within an eight-week period. Athletes’ heart rate was monitored during the SSG. The variables analyzed were the percentage mean heart rate (HR_MEAN%_) and the percentage peak heart rate (HR_PEAK%_). For the analysis of within-session reliability, the mean value of the first two and last two SSG bouts performed within one day were used. The between-session reliability was calculated using the mean value of the four SSG bouts of each SSG type performed on two different days. In both SSGs, the values for reliability were significant and were classified as moderate to excellent. There were no significant differences in the physiological demand among SSG types. We concluded that the offside rule does not influence the physiological demand in a three vs. three SSG and the HR_MEAN%_ and HR_PEAK%_ present moderate to excellent reliability in a three vs. three SSG with and without the offside rule.

## 1. Introduction

In recent years, the physical conditioning of soccer players has developed according to an integrated approach involving tactical and technical aspects of the game [1,2]. In this context, small-sided games (SSGs) provide high-intensity activity, including both tactical and technical demands, and optimize the available training time [3]. Knowledge of the effect of changing SSG characteristics (e.g., the number of players per team, the pitch size, and the rules) helps strength and conditioning coaches to adequately prescribe an SSG during the training process [4]. Although there are many studies on the effect of changing the pitch size and the number of players in a team [5,6,7], there has been less research on how rule changes in SSGs affect the players’ physical and physiological responses [8,9,10,11]. One task constraint that can induce changes in players’ available space is the offside rule, as it might reduce the effective playing area when the defending team moves towards the opponent’s goal. To the best of our knowledge, the influence of the offside rule on the physiological demands of SSGs has not been investigated. Considering the importance of this rule on the game dynamics and the possibility of implementing it during game-based tasks such as an SSG, it is essential to understand its impact on athletes’ physiological responses.

The players’ movements and displacements during official matches are determined by the effective playing area, which is influenced by the offside rule. This constraint causes the playing area to be dynamic, restricting or allowing players to move across the field length according to the position of teammates and opponents [12]. Hence, the relative area (i.e., area per player) also constantly changes during the game [13]. Previous studies have suggested reducing the relative area by decreasing the absolute pitch size (area in m^2^) while maintaining the number of players [14], or keeping the absolute pitch size and increasing the number of players [15]. A smaller relative area generally decreases players’ physical [9,16] and physiological [4] responses, because it constrains players’ displacements. Therefore, another possibility to modulate the relative area in SSGs is the inclusion of the offside rule, because it can reduce the effective space in which players can move. However, some studies on soccer SSGs have included the offside rule [17,18], while others have not [19,20]. Castillo et al. [9] compared the physical demands of a soccer SSG with and without the offside rule and found a greater total distance and a larger distance covered between 13 and 16 km/h on the pitch without the offside rule. Therefore, it might be expected that non-offside SSGs lead to greater physiological responses from the players. Nonetheless, the influence of this rule on the relative area and consequently on the physical and physiological demands of soccer SSGs requires deeper investigation. Moreover, this knowledge may add a new interpretation to previous studies on SSGs that have or have not implemented the offside rule. Understanding the impact of this rule on athletes’ responses can help the coaches to better use game-based activities during training.

Another critical issue regarding the use of SSGs is their reliability as a means of training. This analysis is crucial to test whether specific demands can be achieved when an SSG format is repeatedly applied during the training process. Weir [21] suggested using the intraclass correlation coefficient (ICC) and the standard error of the measurement (SEM) to analyze the reliability. The ICC provides information on the variability between individuals and the consistency of this variability in repeated test measures [21], while the SEM reflects the degree of fluctuation of the individual’s scores in a test or condition, representing the expected natural variability (the random error) for that score [21]. Some studies have investigated the reliability of the physiological responses during different soccer SSGs and presented within- [20] and between-session designs [22]. Many of these studies showed high reliability for physiological demands [22,23,24,25,26,27,28]. A recent systematic review indicated that internal loads—average heart rate (%HRavg), peak heart rate (%HRpeak), and maximum heart rate (%HRmax)—showed small within-session variations (~0.5–6% of change between the lowest and the highest sets/repetitions), irrespective to the SSG format. Therefore, it is possible to expect high reliability of internal load measures in both with and without offside SSGs in the current study [29].

Considering these issues, this study aimed to (i) compare the physiological demands of a three vs. three SSG with and without the offside rule and (ii) to verify the within- and between-session reliability in these two SSGs.

## 2. Materials and Methods

### 2.1. Participants

Twenty-four U-17 male soccer athletes (age: 16.7 ± 0.6 years; body mass: 64.8 ± 6.7 kg; height: 176.5 ± 6.5 cm; body fat: 9.7 ± 1.6%; and estimated VO_2MAX_: 52.1 ± 2.5 mL·kg^−1^·min^−1^) from an elite club participated in this study. This club was considered elite as players compete at the national level regularly. The club achieved first position in the national U-18 competition in the same year the data collection was performed. The athletes competed at a national level and had seven training sessions per week. Data from two athletes were excluded from the analyses due to technical problems, which reduced the final sample to twenty-two players. Players were included if they volunteered to participate in the study and were not injured or returning from injury. On the other hand, the exclusion criteria comprised being injured, not participating in the whole data collection, or refusing to provide written consent to participate in the study. Goalkeepers participated in the data collection but were not evaluated. The participants and their legal guardians were informed about all the research procedures and provided written consent for participating in the study. The local Ethics Committee from the Universidade Federal de Minas Gerais (70103017.0.0000.5149) approved the study, and all the guidelines from the Declaration of Helsinki were followed.

### 2.2. Teams’ Composition for the SSG

The 24 athletes were randomly allocated into eight teams of three players (A to H). Each team had a defender, a midfielder, and a forward to allow teams to explore the physical, technical, and tactical specificities of each playing position during the different SSGs [30,31]. The eight teams were divided into two groups. Group 1 was composed of teams A to D, and Group 2 was composed of teams E to H. Each team within the group played against the same opponent during the entire study (e.g., Team A always played against Team B) to reduce the possible variability related to differences in the opposing teams during the SSGs [32]. The procedures for the composition of the teams and groups are described in Figure 1.

### 2.3. Data Collection

Athletes performed several 3 vs. 3 SSGs (where goalkeepers were included but not evaluated) with (3vs.3_WITH_) and without (3vs.3_WITHOUT_) the offside rule. Both of the SSGs were played in the 3 vs. 3 format, on a 36 × 27 m pitch of natural grass, with goals measuring 6 × 2 m (see Figure 2). In the 3vs.3_WITH_ game, two referees were positioned on the sides of the pitch to observe the game and apply the offside rule when necessary. The defending team received a free kick when an offside situation was detected. In the 3vs.3_WITHOUT_ game, the offside rule was not applied, so players could play freely. Each session comprised four SSG bouts, which lasted for four minutes, with five minutes of passive rest. Additional balls were placed around the pitch to ensure a fast game restart when the ball went off the pitch. Coaches and researchers did not give the players verbal encouragement or technical instructions.

The SSGs were performed on Tuesdays and Wednesdays for eight consecutive weeks at the end of the competitive season. We chose the same weekdays to minimize the influence of the distribution of training loads on athletes’ physical responses. Group 1 performed the SSGs during the first four weeks (one SSG format each day), while Group 2 performed the SSGs in the last four weeks. This was to avoid a long break between SSG sessions for each team, which could lead to changes in physical fitness, and to minimize the disruption to the athletes’ training routines. Therefore, each SSG format was performed twice, with an interval of six to eight days between trials for each SSG format, according to the club availability.

To standardize the influence of circadian rhythm on the observed responses, all sessions were performed at the same time (between 8 a.m. and 10:30 a.m.). The mean (the standard deviation) temperature and relative humidity of all sessions were 31.1 °C (± 2.6 °C) and 28.1% (± 4.8%), respectively, recorded by a portable digital thermometer (Big Digit Hygro-Thermometer, Extech Instruments, Massachusetts, EUA).

To control for the possible effect of changes in physical conditioning on the reliability analysis, athletes performed the Yoyo Intermittent Recovery Test Level 1 (Yo-YoIR1) [31] and a 20 m sprint test one week before and two weeks after the data collection.

In detail, the protocol used for the 20 m sprint test consisted of taking four attempts at the 20 m test, and time recording the distance covered. An interval of three minutes of passive recovery between attempts was established. It is noteworthy that the distance of 20 m was chosen for the measurement of running speed due to evidence that, in official games, sprint running distances longer than 20 m are infrequent [33].

The Yo-YoIR1, on the other hand, is an intermittent, progressive aerobic capacity test, in which athletes perform a series of round-trip runs on a 20 m course [31]. So after each round trip, there is an interval of 10 seconds of active rest in which the athlete trots or walks a course of 10 m, covering 5 m going and 5 m returning. The running speed is determined by sound signals, starting at 10 km/h and increasing progressively throughout the test. In the present study, when the athlete was unable to maintain the rhythm (the speed) determined by the sound signals for two consecutive series, the test was closed, and the total distance covered was recorded. The peak heart rate achieved during Yo-YoIR1 was considered as the athletes’ maximum heart rate and was used to relativize heart rate values as a percentage of the maximum.

### 2.4. Physiological Demand

The heart rate (HR) of the players during the SSGs was recorded using a 1 Hz heart rate monitor (Polar T31 Electro Oy^®^, Kempele, Finland). The reliability of this device has been previously tested in the literature. Physiological demands were characterized by the percentage of mean heart rate (HR_MEAN%_) and the percentage of peak heart rate (HR_PEAK%_). The HR_MEAN%_ was calculated as the mean of all the values recorded by HR monitors during the SSG bouts (HR values of the rest intervals were excluded). The HR_PEAK%_ was considered to be the highest value recorded during the SSG bouts. All HR values were relativized by the peak HR presented by each athlete in the Yo-YoIR1.

### 2.5. Statistical Analyses

The data did not present significant deviations from normality (using Shapiro–Wilk’s test) or homoscedasticity (using Levene’s test). An independent t-test was used to compare means between the 3vs.3_WITH_ and 3vs.3_WITHOUT_ games. Cohen’s d effect size was calculated to characterize the magnitude of the significant differences in paired comparisons and was classified as insignificant (<0.19), small (0.20–0.49), medium (0.50–0.79), or large (≥0.80) [32].

For the within-session reliability of the HR_MEAN%_ and HR_PEAK%_ for the 3vs.3_WITH_ and 3vs.3_WITHOUT_ games, athletes’ mean values of the first two and the last two SSG bouts in each session (day 1 and day 2) were used. To determine the between-session reliability, athletes’ mean values of the four SSG bouts performed in each session were used. For both within- and between-session reliability, the intraclass correlation coefficient 2,k (ICC_2,k_) and the standard error of the measurement (SEM) were used [21]. The ICC_2,k_ values were classified as weak (<0.4), moderate (0.40–0.59), good (0.60–0.74), or excellent (0.75–1.00) [34].

A two-way analysis of variance (groups × moments) was used to compare the data on aerobic power (from the Yo-YoIR1) and sprint performance (from the 20 m sprint) among the two groups and moments (from the pre- and post-data collection).

The level of statistical significance was set at 5% (α = 0.05). All analyses were performed using SPSS version 23.0 (Chicago, IL, USA).

## 3. Results

Table 1 shows the descriptive data (the means and standard deviations) of HR_MEAN%_ and HR_PEAK%_ in the investigated SSGs. There were no significant differences between the SSGs with and without the offside rule (giving a small effect size).

Table 2 shows the within-session (bouts within days 1 and 2) intraclass correlation coefficient values (95% CI), the ICC classification, and the SEM values for the variables related to the physiological demand of SSGs with and without the offside rule. The ICC values were classified as “good” or “excellent” (values above 0.60), except for the HR_MEAN%_, which was classified as “moderate” on day 2.

Table 3 shows the between-session (between days 1 and 2) intraclass correlation coefficient values (95% CI), the ICC classification, and the SEM values for the variables related to the physiological demand of SSGs with and without the offside rule. The ICC values were classified as “good” or “excellent” (values above 0.60), except for the HR_MEAN%_ in the 3vs.3_WITH_ game, which was classified as “moderate”.

The two way analysis of variance of the control variables (aerobic power—pre-test: 1850.9 ± 288.7 m; post-test: 1950.0 ± 277.6 m and 20 m sprint performance—pre-test: 22.7 ± 0.6 km/h; post-test: 23.3 ± 0.6 km/h) showed no significant interaction (aerobic power—F = 0.68; *p* = 0.41; 20-m sprint performance—F = 0.985; *p* = 0.325) or main effects (aerobic power—F = 3.47; *p* = 0.07; 20 m sprint performance—F = 0.352; *p* = 0.556). These data show the lack of differences in physical conditioning during the period of the data collection, mitigating the possible effect of variability on the between-session reliability analysis.

## 4. Discussion

This study aimed to investigate the effect of the offside rule on the physiological demands of three vs. three soccer SSGs in U-17s and the reliability of the physiological demands in three vs. three SSGs with and without the offside rule. The results show that the physiological demands, characterized by the HR_PEAK%_ and HR_MEAN%_, did not differ among the SSGs with and without the offside rule, and thus, our hypothesis was rejected. Furthermore, the within and between-session reliability of physiological demands confirmed our hypothesis, with moderate to excellent ICC values for all variables, regardless of the rules of the SSGs.

We expected that the offside rule would decrease the physiological demands of the SSGs because of the reduction in the effective playing area. This hypothesis was based on previous results that showed a decrease in the physiological demands when the absolute playing area was decreased for the same number of players or when the number of players was increased within the same playing area [16]. These changes result in smaller relative areas (i.e., area per player), restricting the available space for players to move around and, consequently, reducing the intensity of the game (i.e., lower physiological demands) [14,35]. A previous systematic review included studies with similar playing areas and showed that reducing the relative area per player tends to reduce physical and physiological responses [16]. However, the results of the present study do not corroborate this hypothesis. A possible explanation for these divergent results may be related to the magnitude of the change in the effective playing area in the three vs. three SSG with the offside rule. Previous studies have shown that small changes in the relative area may not be sufficient to influence players’ physical [9] and physiological [4] responses. Specifically, the reduction in the effective playing area depends on the defending team moving up the pitch to constrain the available space for the offensive team. Therefore, the number of times the players adopted this behavior might have been smaller than what was required to induce different responses when considering the whole bout. Moreover, although the heart rate has been widely used in studies on SSGs [36] and is considered a valid variable to measure SSG intensity [37], it may not be sensitive enough to detect differences in the frequency of specific actions (i.e., jumps, duels, accelerations, decelerations, sprints, and changes of direction) during the game, which could, in turn, also reflect game intensity [38]. Considering this issue, future studies should collect information through other variables, such as accelerations, decelerations, mean speed, and distances covered in different speed zones, to increase the understanding of exercise intensity during game-based activities, such as SSGs [39].

The HR_PEAK%_ and HR_MEAN%_ values found in both SSGs investigated in this study are similar to those reported in previous studies on the three vs. three SSG format performed by soccer players of a similar age (Sub-17) [40,41,42]. Furthermore, studies on SSG training (training periods above four weeks) indicate the necessity for HR_mean_ values to be above 80% of HR_max_ to improve aerobic performance [43,44,45,46]. Therefore, the results of the present study reinforce the potential use of different SSGs for the improvement of aerobic performance in soccer athletes, including the offside rule.

The investigation of SSG reliability is essential to support using SSGs during training. In addition, with the knowledge of the demands imposed on athletes by different SSGs, strength and conditioning coaches can examine if those demands are reproducible when the same SSG is performed at different moments. In the present study, high ICC (>0.60) and low SEM (<1.7) values were found in the within-session reliability analysis of HR_MEAN_. These data corroborate the results of previous studies on the reliability of heart rate variables collected during SSGs, despite the differences in the SSG formats. Hill-Haas et al. [24] compared different SSG formats (two vs. two, four vs. four, and six vs. six) and found percentage values of SEM (SEM%) of 1.9 and 4.4% for the HR_PEAK%_ and 1.1 and 3.6% for the HR_MEAN%_. Another study also reported small SEM percentage values for the HR_MEAN%_ (5.4%) and HR_PEAK%_ (3.0%) in a three vs. three SSG with similar characteristics [22]. Finally, Stevens et al. [28] found good reliability values for the HR_MEAN%_ during a six vs. six SSG (ICC = 0.61 and SEM% = 2.2%). On the other hand, the results of the present study on between-session reliability suggest good reproducibility of the HR_PEAK%_ and HR_MEAN%,_ despite an interval of one week between the sessions (ICC = 0.56), with a low variability among these measures (SEM < 2.6%). These results are similar to previous research that indicated good reproducibility for the physiological demands represented by heart rate variables in different SSGs. Da Silva et al. [23] and Rampinini et al. [27] investigated the reliability of the HR_MEAN%_ in SSGs with different numbers of players and pitch sizes and found that values of SEM percentage ranged from 2.2 and 3.4%, and the percentage of typical error (TE%) values (similar to SEM) ranged from 2.0% and 5.4%, respectively. Additionally, Hill-Haas et al. [25] found low variability for the HR_MEAN%_, with TE% values ranging from 2 and 4% in a four vs. four SSG. This result is similar to that found by Hulka et al. [47], which showed high ICC (0.88) and low SEM% (2.35%) values in a four vs. four SSG. Additionally, both with and without the offside rule, SSGs showed similar classifications regarding the reliability measures. However, when looking at both within- and between-session reliability, the SSG without the offside rule showed lower ICC values than the SSG with it. It has been proposed in the literature that a higher movement variability can be detected in lesser-known game formats [48,49,50]. It can be argued that U-17 soccer players usually engage in more specific tasks than those that are general game-based tasks—therefore, the game with the offside rule seems to be more representative of the requirements of the official match. Consequently, the reduction in the reliability might indicate a more variable displacement behavior in the SSGwithout condition due to the players’ need to readapt to the new constraints.

This study investigated U-17 athletes, which hinders the generalization of the results to other age categories. Future studies should be carried out with athletes of different ages to provide more precise information on the physiological demands of the three vs. three SSGs investigated in this study. Moreover, this study did not monitor athletes’ recovery levels during the data collection, which could have added a deeper understanding of athletes’ conditions while recording the variables. In this case, further research should investigate athletes’ recovery behavior over SSG bouts and between training sessions to provide information that better supports the use of SSGs for the physical conditioning of soccer players.

## 5. Conclusions

Using the offside rule in a three vs. three SSG did not influence the physiological responses of young soccer athletes. The within- and between-session reliability values of the physiological variables in both SSGs with and without the offside rule were high, supporting the reproducibility of the physiological demands of SSGs despite their natural unpredictability and variability. The absence of difference between the protocols indicates that coaches might choose between the two SSG formats based on other goals—for example, tactical missions related to enlarging the surface area—instead of considering the impact the offside rule will have on players’ physiological responses.

## Figures and Tables

**Figure 1 ijerph-19-10544-f001:**
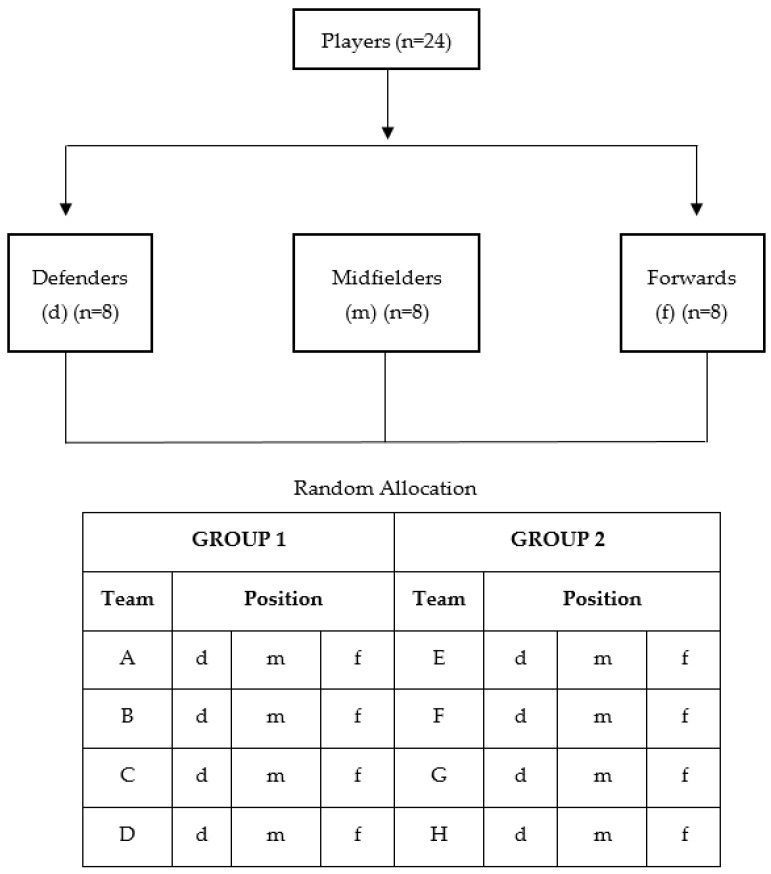
Team and group composition procedures. Legend: d = defender; m = midfielder; f = forward.

**Figure 2 ijerph-19-10544-f002:**
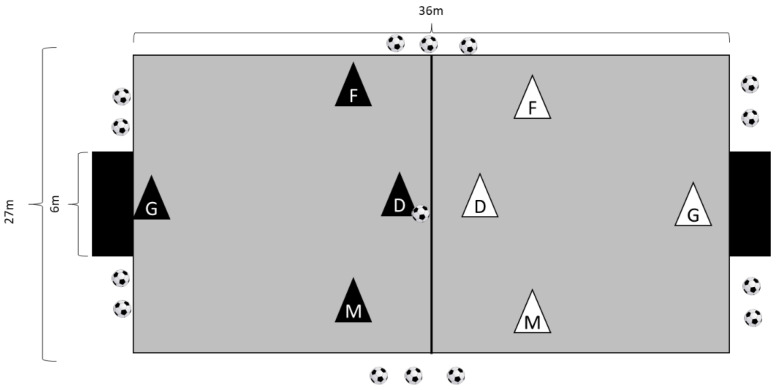
Representation of the 3vs.3_WITHOUT_ game. Legend: G = goalkeeper; D = defender; M = midfielder; F = forward.

**Table 1 ijerph-19-10544-t001:** Means (standard deviations) of the variables related to the physiological demand of SSGs with and without the offside rule.

	3vs.3_WITH_	3vs.3_WITHOUT_			
	Mean (SD)	Mean (SD)	*p*-Value	ES	Interpretation
**HR_PEAK%_**	94.8 (2.1)	94.4 (2.1)	0.14	0.33	Small
**HR_MEAN%_**	87.4 (2.9)	87.1 (2.3)	0.46	0.16	Insignificant

Legend: **3vs.3_WITH_** = small-sided games with the offside rule; **3vs.3_WITHOUT_** = small-sided games without the offside rule; **FC_PEAK%_** = percentage peak heart rate; **FC_MEAN%_** = percentage mean heart rate.

**Table 2 ijerph-19-10544-t002:** Within-session intraclass correlation coefficients (95% CI), ICC classification, and SEM for the variables related to the physiological demand of SSGs with and without the offside rule.

	HR_PEAK%_	HR_MEAN%_	HR_PEAK%_	HR_MEAN%_
	3vs.3_WITH_—DAY 1		3vs.3_WITH_—DAY 2	
**ICC (95% CI)**	0.76 * (0.36–0.91)	0.85 * (0.44–0.95)	0.75 * (0.12–0.91)	0.73 * (0.32–0.89)
**ICC Classification**	Excellent	Excellent	Excellent	Good
**SEM (%)**	1.3	1.7	1.4	2.1
	**3vs.3_WITHOUT_—DAY 1**		**3vs.3_WITHOUT_—DAY 2**	
**ICC (95% CI)**	0.61 * (−0.23–0.87)	0.62 * (−0.21–0.87)	0.73 * (−0.20–0.93)	0.58 * (−0.21–0.87)
**ICC Classification**	Good	Good	Good	Moderate
**SEM (%)**	1.3	1.7	1.0	1.3

**3vs.3_WITH_** = small-sided games with the offside rule; **3vs.3_WITHOUT_** = small-sided games without the offside rule; **FC_PEAK%_** = percentage peak heart rate; **FC_MEAN%_** = percentage mean heart rate; **CI** = confidence interval; **SEM** = standard error of the measurement. ***** indicates statistical significance (*p* < 0.05).

**Table 3 ijerph-19-10544-t003:** Between-session intraclass correlation coefficients (95% CI), ICC classification, and SEM for the variables related to the physiological demand of SSGs with and without the offside rule.

	3vs.3_WITH_		3vs.3_WITHOUT_	
	HR_PEAK%_	HR_MEAN%_	HR_PEAK%_	HR_MEAN%_
**ICC** **(95% CI)**	0.62 * (0.09–0.85)	0.56 * (−0.04–0.82)	0.77 * (0.42–0.91)	0.69 * (0.25–0.88)
**ICC Classification**	Good	Moderate	Excellent	Good
**SEM (%)**	1.8	2.6	1.4	1.8

**3vs.3_WITH_** = small-sided games with the offside rule; **3vs.3_WITHOUT_** = small-sided games without the offside rule; **FC_PEAK%_** = percentage peak heart rate; **FC_MEAN%_** = percentage mean heart rate; **CI** = confidence interval; **SEM** = standard error of the measurement. ***** indicates statistical significance (*p* < 0.05).

## Data Availability

Not applicable.

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
