# Peer review of "Effect of Small-Sided Games with and without the Offside Rule on Young Soccer Players: Reliability of Physiological Demands"

_ijerph, 2022, doi:10.3390/ijerph191710544_

Round 1

Reviewer 1 Report

Thank you for the opportunity to review this work. The manuscript under this review compares the physiological demand between 3vs.3 small-sided games with and without the offside rule, as well as the within and between-sessions reliability of this demand.

The study is interesting and original with various minor considerations before the manuscript could be considered for publication in Int. J. Environ. Res. Public Health.

Abstract

The presentation of the abstract is totally clear.

Lines 17-19: avoid repeating "for the analysis" in successive sentences.

Lines 20-22: I suggest that the authors change the order of the two sentences, presenting the significant results first and then the non-statistically significant results.

Introduction

I congratulate the authors for a well-written introduction with a good bibliography. This section is well written and referenced. Clear and orderly with a well-executed theoretical framework.

Lines 67-70: I suggest not to explain the meaning of ICC and SEM, in the reference of Weir's study it is defined and there is no need to repeat it in the manuscript.

Line 74: the first time the meaning of the acronym heart rate is used, explain it as "heart rate (HR)".

The objectives are clear and well defined by the authors.

Materials and Methods

Line 83: 2.1. Participants: remove bold and present it in italics like the rest of the subsections.

Line 89: What ethics committee? This needs to be stated in the manuscript as well as in the Institutional Review Board Statement section with the assignment code.

Line 103: 2.2 Data Collection; please change it to "2.3. Data collection" and renumber the following subsections of Materials and Methods.

Line 136: please explain the acronym for peak heart rate the first time it is presented in the manuscript (HRPEAK).

Lines 147-148: Please submit this exclusion sentence for players in the subsection of 2.1. Participants, justifying that they undertook the entire data collection and research process, but were excluded in the analysis.

Results

Table 1. In the legend add the explanation of SD (standard deviation) and remove the explanation of p, placing inside the table for "p-value". Remove the explanation of ES, as it is already presented in the statistical analysis.

Table 2. in the legend, delete the ICC classification, it is already explained previously in the statistical analysis, as well as in the legend of table 3.

Discussion

The discussion is clear, well laid out and with references that justify this section with the results previously found in previous studies, marking the novelty of this research.

The limitations and future studies proposed by the authors are correct.

Conclusions

The conclusions are perfectly in line with the objectives set and the results provided by the authors.

Reviewer 2 Report

Dear Authors

Changes are presented

King Regards

Reviewer 3 Report

Comments to Authors:

General Comments: The authors sought to investigate how manipulation of the rules in small sided games impacted the physiological demands of the game. Overall, the study is well planned and accomplished these goals. At this is time I have a few points of clarification before acceptance for publication.

Introduction:

The premise of the study is that reduction in playing area impacts the physiological demand on the individual players. What was the size of the playing areas in the previous studies? I believe based on the results of this study and how those findings deviant from previous findings is an interesting note in explaining your findings.

Discussion

As mentioned previously, the size of the playing area could have an influence on the findings of this study. Even with the offside rules in place no difference where seen between conditions. Could this simply be an outcome of a large area with few players.  Regards of the rule there was a large area to cover those physiological demand was high. Without any external load metric (total yardage or high speed running) it is hard to say that the offside rules truly influence the displacements of each individual. Again this could be added to help in explaining the results since the do differ from previous investigations.

I feel it is also important to offer reasoning as to why the ICC between sessions values differ between conditions. It appears that the 3v3with values have limited reliability. The drop from excellecnt to good for HRpeak% appears to be an issue and again could be used in explaining the how differences were not seen between conditions.

Round 2

Reviewer 2 Report

Accept in present form